# IL-1 Receptor Contributes to the Maintenance of the Intestinal Barrier via IL-22 during Obesity and Metabolic Syndrome in Experimental Model

**DOI:** 10.3390/microorganisms12081717

**Published:** 2024-08-20

**Authors:** Melissa S. G. Machado, Vanessa F. Rodrigues, Sara C. Barbosa, Jefferson Elias-Oliveira, Ítalo S. Pereira, Jéssica A. Pereira, Thaílla C. F. Pacheco, Daniela Carlos

**Affiliations:** Laboratory of Immunoregulation of Metabolic Disease, Ribeirão Preto Medical School, University of São Paulo, Ribeirão Preto 14049-900, SP, Brazil; melissa.santana04@usp.br (M.S.G.M.); vanessa.fr@usp.br (V.F.R.); saracandidab@gmail.com (S.C.B.); jeffersonelias@usp.br (J.E.-O.); italo.biotecnologista@gmail.com (Í.S.P.); jessicaassis@usp.br (J.A.P.); thaillapacheco@usp.br (T.C.F.P.)

**Keywords:** IL-1R1, obesity, metabolic syndrome, IL-22, ILC3, gut permeability, bacterial translocation

## Abstract

Intestinal permeability and bacterial translocation are increased in obesity and metabolic syndrome (MS). ILC3 cells contribute to the integrity of intestinal epithelium by producing IL-22 via IL-1β and IL-23. This study investigates the role of IL-1R1 in inducing ILC3 cells and conferring protection during obesity and MS. For this purpose, C57BL/6 wild-type (WT) and IL-1R1-deficient mice were fed a standard diet (SD) or high-fat diet (HFD) for 16 weeks. Weight and blood glucose levels were monitored, and adipose tissue and blood samples were collected to evaluate obesity and metabolic parameters. The small intestine was collected to assess immunological and junction protein parameters through flow cytometry and RT-PCR, respectively. The intestinal permeability was analyzed using the FITC-dextran assay. The composition of the gut microbiota was also analyzed by qPCR. We found that IL-1R1 deficiency exacerbates MS in HFD-fed mice, increasing body fat and promoting glucose intolerance. A worsening of MS in IL-1R1-deficient mice was associated with a reduction in the ILC3 population in the small intestine. In addition, we found decreased IL-22 expression, increased intestinal permeability and bacterial translocation to the visceral adipose tissue of these mice compared to WT mice. Thus, the IL-1R1 receptor plays a critical role in controlling intestinal homeostasis and obesity-induced MS, possibly through the differentiation or activation of IL-22-secreting ILC3s.

## 1. Introduction

Obesity, as defined by the World Health Organization (WHO), is an excess of body fat in an amount that determines health impairments and is identified as a risk factor for various chronic diseases, such as cardiovascular diseases, type 2 diabetes mellitus (T2DM), chronic kidney disease, hepatic steatosis, and a range of musculoskeletal disorders [1]. It has been discovered that in cases of obesity and T2DM, there is an increase in the plasma levels of pathogen-associated molecular patterns (PAMPs) such as lipopolysaccharide (LPS), a process called endotoxemia, as well as bacterial translocation into adipose tissue [2]. Both endotoxemia and bacterial translocation occur due to alterations in the integrity of the intestinal barrier, with the reduced expression of tight junction proteins (TJs), such as ZO-1, occludins, and claudins, leading to increased intestinal permeability and allowing the passage of these bacterial products [2,3].

Several factors work together to enable the intestine to absorb nutrients while simultaneously preventing the passage of harmful elements into the bloodstream [4]. In this regard, the immune system plays a significant role, as it represents strategically located leukocytes between epithelial cells, in the lamina propria, and in lymphatic structures. Among the main cells in this system are type 3 innate lymphoid cells (ILC3), predominantly found in the intestine, which promote the proliferation of intestinal epithelial cells, strengthen the epithelium, and induce the gene expression of antimicrobial peptides (AMPs), such as Reg3γ, restoring intestinal barrier function and controlling local microorganisms [5]. These cells predominantly reside in the intestinal mucosa, limiting gut microbiota and pathogenic microorganisms. ILC3s, along with ILC1s and ILC2s, are part of a group of ILCs that have specific transcription factors and produce different cytokine profiles resembling helper T lymphocytes, which account for adaptive immunity (Th1, Th2, and Th17). However, unlike T lymphocytes, ILCs do not express TCR, the receptor used for antigen recognition; therefore, they have early action in response to local cytokines [6].

ILC3s express the RORγt transcription factor to promote the production of IL-22, as well as IL-17, through the stimulation of IL-1β and IL-23 produced by local dendritic cells and macrophages [7]. The cytokines IL-17 and IL-22, produced by ILC3, confer protection against bacterial and fungal infections and contribute to the shape of gut microbiota. Additionally, IL-22 can promote the expression of TJ proteins, such as occludins, claudins, and ZO-1, which are important for maintaining the tight adherence of epithelial cells and preventing intestinal barrier permeability [8,9]. In this context, researchers have shown that IL-1β and IL-23 act synergistically to induce the production of IL-22 by ILC3 cells in intestinal bacterial infections [10]. However, there are still no reports on the role of IL-1 in the response of ILC3 cells in the context of obesity and MS. Therefore, in this work, we aim to investigate the role of the IL-1R1 receptor in inducing ILC3 cells and its implication in obesity and MS. Based on this evidence, our hypothesis is that IL-1, via IL-1R1, promotes the induction of intestinal ILC3 cells and reinforces the intestinal barrier through of the production of IL-22, as well as the improvement of metabolic and inflammatory changes associated with obesity and MS.

## 2. Material and Methods

### 2.1. Animals

Female, 6-week-old, C57BL/6 wild-type (WT) and IL-1R1 knockout (KO) mice were used. Mice were kept in the animal house of the Department of Biochemistry and Immunology, FMRP-USP, Ribeirão Preto, where they were provided with filtered air and with water and food ad libitum. All procedures were carried out following the principles proposed by the National Council for the Control of Animal Experimentation (CONCEA) and were approved by the Ethics Committee on Animal Use (CEUA) of the Ribeirão Preto Medical School at the University of São Paulo, protocol 53/2021.

### 2.2. Experimental Groups and Obesity Induction

The mice were divided into 4 groups of 4 to 6 mice each: group I, WT mice fed a standard control diet (SD) (AIN93 M—composed of 10% fat, 70% carbohydrate, and 20% protein); group II, WT mice fed a high-fat diet (HFD) (D12492—composed of 60% fat, 20% carbohydrate, and 20% protein—PRAGSOLUÇÕES^®^); group III, IL-1R1^−/−^ mice fed an SD; and group IV, IL-1R1^−/−^ mice fed an HFD. All animals received 4 g of food daily for 16 weeks.

### 2.3. Nutritional and Obesity Parameters

The nutritional profile was determined by the analysis of food and caloric ingestion, body weight gain, visceral fat depot, total body fat, and adiposity index. The total weight was measured weekly using a digital scale. The amount of total fat mass was determined by the sum of the retroperitoneal and mesenteric fat deposits. The adiposity index, used to determine possible obesity, was calculated by dividing the total body fat by the final body weight, multiplied by 100.

### 2.4. Assessment of Glucose Intolerance

Mice were fasted for 6 h, and blood was collected from the tip of the tail vein to assess the fasting blood glucose. Also, during the 16th week of the experiment, mice were fasted for 12 h for the glucose tolerance test (GTT). For the GTT, blood samples were taken at baseline and after intraperitoneal administration of glucose 25% (Sigma^®^-Aldrich , San Luis, MO, USA) equivalent to 2.0 g/kg and were collected at 0, 15, 30, 60, and 120 min. The ACCU-CHEK^®^Active glucometer (ROCHE, Basel, Switzerland) was used to read the blood glucose levels.

### 2.5. Detection of Total Cholesterol

After the euthanasia, blood was collected by cardiac puncture. Hemolysis-free serum was collected after centrifugation. The total cholesterol concentration was measured using a kit from Labtest (St. Louis, MO, USA), following the manufacturer’s instructions. 

### 2.6. RNA Extraction and Analysis of Gene Expression by RT-qPCR

The extraction of total RNA from the ileum samples was performed using the Promega RNA extraction kit following the manufacturer’s recommendation. The RNA samples were suspended in 50 µL of nuclease-free deionized water and stored at −80 °C. The RNA concentration was determined using Nanodrop 2000 device (Thermo Fisher Scientific, Waltham, MA, USA). Then, the complementary DNA was constructed using the High-Capacity cDNA Reverse Transcription Kit (Thermo Fisher Scientific, Waltham, MA, USA), Oligo(dT)23 Anchored (Sigma-Aldrich, San Luis, MO, USA), and a Mastercycler (Eppendorf, Wesseling-Berzdorf, Germany). The quantitative gene expression of zonula occludens (ZO-1), Reg3γ, IL-17A, and IL-22 were analyzed through qPCR reactions using SYBR Green at a Quantistudio 1 (Thermo Fisher Scientific, Waltham, MA, USA) equipment. The basic amplification reaction comprised 2 min at 50 °C, 10 min at 95 °C, forty cycles of 15 s each at 95 °C, and 1 min at 60 °C, followed by a final cycle of 20 min, with a temperature ramp from 60 to 95 °C. The results were analyzed based on CT (cycle threshold) value.

### 2.7. DNA Extraction and Analysis of Bacterial Genes by qPCR

DNA extraction from the visceral adipose tissue (VAT) was performed using the DNeasy Blood and Tissue kit from QIAGEN (Hilden, Germany) following the manufacturer’s recommendations. For the qPCR analysis, 10 ng of DNA and 5 μM of bacterial 16S primes were used. For the phylum and specie determination of gut microbiota, the amplifications were carried out using StepOne equipment (Applied Biosystems, Waltham, MA, USA). The DNA from feces collected on 16th week was extracted using the DNeasy PowerSoil Pro kit (Qiagen, Hilden, German) following the manufacturer’s instructions. Also, DNA extraction from the VAT was performed using the DNeasy Blood and Tissue kit from QIAGEN. The DNA extracted was quantified using the NanoDrop 2000 (Thermo Fisher Scientific, Waltham, MA, USA). Analysis of the relative abundance of the main bacterial phyla Bacillota, Bacteroidota, Actinobacteriota, Verrucomicrobiota, Pseudomonadota, Mycoplasmatota, and Akkermansia muciniphila, Escherichia coli, Clostridium leptum and Ruminococcus torques species was determined using 10 ng of DNA and 5 μM of primers (Appendix A), SYBR Green Master Mix (Promega, Madison, WI, USA) and StepOne qPCR (Life Technologies, Carlsbad, CA, USA).

### 2.8. Intestinal Permeability by FITC-Dextran

After 12 h of fasting, mice received fluorescein isothiocyanate (FITC)-dextran by gavage (400 mg/kg) (Sigma-Aldrich, San Luis, MO, USA). After 4 h, the blood samples were collected from the tail vein tip. The blood was centrifuged at 4 °C, 10,000× *g* for 2 min. The serum was then collected, and samples of FITC-Dextran were diluted in phosphate-buffered saline (PBS) to use as a standard curve. The absorbance of 50 µL diluted serum at 1:5 or the standard curve (0.78–100 µg/mL range) was measured in a FlexStation (Molecular Devices, San José, CA, USA) with excitation at 485 nm and reading at 535 nm.

### 2.9. Cell Extraction from Lamina Propria

The cells from lamina propria were isolated from the digestion of the small intestine after the Peyer’s patches and feces were removed. The small intestine was fragmented into 2-cm segments and incubated under agitation at 37 °C in 20 mL of Hanks’ balanced salt solution (HBSS) containing 10 mM de HEPES (Sigma-Aldrich, San Luis, MO, USA), 2% fetal bovine serum (FBS), and 5 mM EDTA. After incubation, the segments were transferred to a tube containing 3 mL of incomplete HBSS medium supplemented with 10 U/mL of DNAse (Sigma) and 1 mg/mL of IV collagenase (Sigma) and incubated under agitation at 37 °C for 30 min. Next, 20 mL of complete RPMI medium with 3% FBS was added to inhibit the digestion enzyme activity. The material was mashed through a 70 µm cell strainer. After centrifugation (1500 rpm, 10 min at 4 °C), the supernatant was discarded, and 10 mL of complete RPMI with 3% of FBS was added. The material was transferred and filtered through a 40 µm cell strainer and centrifuged. The supernatant was discarded, and the cells were marked for flow cytometry analysis.

### 2.10. Analysis of Leukocytes by Flow Cytometry

The flow cytometer analysis was performed on lamina propria cell samples with 1 × 10^6^ cells/tube in 50 µL of PBS. After 15 min of incubation with the 2.4G2 cell culture supernatant, cells were incubated for 30 min with the following anti-mouse antibodies for external staining: anti-CD45 PECy7, CD3 efluor 450, CD90.2 APC-Cy7, Lin FITC, and CD4 APC. Next, cells were permeabilized using a custom kit following the manufacturer’s instruction ( Thermo Fisher Scientific, Waltham, MA, USA). The cells were then internally stained with the anti-mouse antibodies anti-ROR-γt Percp Efluor 710 for 30 min. Cells were analyzed using a FACS Canto flow cytometer, and the data were analyzed using FLOWJO software version 10.9.0 (BD Bioscience, Franklin Lakes, NJ, USA).

### 2.11. Histological Analysis and Calculation of Adipocyte Area

A portion of the VAT was removed and fixed in PBS/10% formaldehyde for histopathological analysis. Following fixation, tissue samples were dehydrated, embedded in paraffin, sectioned, and mounted on glass slides. Slides were then incubated at 60 °C for 1 h. Hydration and deparaffinization were achieved using xylene, alcohol, and water. The sections were stained with hematoxylin and eosin. Microscope images of the tissue were captured at 10× magnification, with 1–6 images taken per animal in a blinded manner. To calculate adipocyte area, the software ImageJ version 1.50i (National Institutes of Health, USA) was calibrated to the 10× magnification. A total of 10 adipocytes per mouse were randomly selected for analysis.

### 2.12. Statistical Analysis

The data are expressed as mean ± SEM. The differences observed among the several experimental groups were analyzed by one-way or two-way ANOVA tests followed by the parametric Tukey post-test for comparing multiple groups. All analyses were performed using PRISM 8.0 software (GraphPad Software, San Diego, CA, USA). Statistical significance was set at *p* < 0.05.

## 3. Results

### 3.1. Deficiency of IL-1R1 Exacerbates Obesity in the Diet-Induced Obesity Model (DIO)

The WT and IL-1R1^−/−^ mice were fed an SD or HFD for 16 weeks, during which their body weight, food, and caloric intake were measured weekly. The body weight variation from group IL-1R1^−/−^+HFD was significantly higher compared to the WT and IL-1R1^−/−^ groups fed an SD, but not compared to the WT+HFD group (Figure 1A–C). In terms of food intake, the IL-1R1^−/−^+HFD group showed a lower food intake compared to the groups that were fed an SD, while the other groups presented similar results (Figure 1D). Regarding caloric intake, the WT and IL-1R1^−/−^ mice fed an HFD showed a significant increase in caloric intake compared to their respective SD controls (Figure 1E), demonstrating the effectiveness of the HFD in providing a greater amount of energy.

Also, white adipose tissues (retroperitoneal and visceral) were collected and weighed to evaluate the body fat gain and calculate the adiposity index. Compared to the other groups, the IL-1R1^−/−^+HFD group showed a significant increase in body fat accumulation (Figure 1F). The HFD induced obesity in both WT and KO groups; however, the deficiency of IL-1R1 resulted in the exacerbation of obesity, proved by the significant increase in adiposity index (Figure 1G). The deposits of visceral adipose tissue (VAT) were increased in mice fed an HFD compared to the groups fed an SD, with emphasis on the IL-1R1^−/−^+HFD group, which showed a significant increase compared to both WT and IL-1R1^-/−^ groups fed an SD (Figure 1H). Furthermore, the histological analysis of the VAT revealed increased adipocyte hypertrophy in the groups fed an HFD, with a further exacerbation observed in IL-1R1^−/−^ mice (Figure 1I,J).

### 3.2. IL-1R1^−/−^ Mice Became More Glucose Intolerant and Develop MS after HFD Feeding

The IL-1R1^−/−^+HFD group showed elevated fasting blood glucose levels only during the fourth and eighth weeks of the experiment compared to the WT+HFD group (Figure 2A). In the final week of the experiment, both HFD-fed groups demonstrated a significant increase in blood glucose only compared with their respective control groups fed an SD (Figure 2B). The mice were also submitted to the glucose tolerance test (GTT) on the 16th week. In the IL-1R1^−/−^+HFD group, most of the mice had glucose levels between 140–199 mg/dL at 2 h of the test, characterizing a prediabetes state (Figure 2C) and indicating a worsening in glucose tolerance. Although IL-1R1^−/−^+HFD mice had the area under the curve elevated, it was significant only compared with WT+SD or IL-1R1^−/−^+SD (Figure 2D). Both groups fed an HFD showed elevated serum cholesterol compared to the respective control groups (Figure 2E). Also, it is important to report that the IL-1R1^−/−^+HFD group had a significant increase compared to the IL-1R1^−/−^+SD group.

### 3.3. IL-1R1^−/−^ Mice Had Compromised ILC3 Induction in the Small Intestine after HFD Feeding

Given the importance of IL-1 as a pivotal cytokine in the induction of ILC3, it was investigated by a flow cytometer whether these cells were reduced within the mucosa of the small intestine, where the majority of ILC3 cells are concentrated. Both WT and IL-1R1^−/−^ HFD-fed groups exhibited a significant reduction in ILC3 frequency compared to WT+SD group. However, this reduction was more prominent in the IL-1R1^−/−^+HFD group and significant compared to the WT+HFD group (Figure 3A,C). A similar outcome was observed in the absolute number of ILC3 cells, although no difference was observed between WT+HFD and IL-1R1^−/−^+HFD (Figure 3B).

Th17 cells share the same transcription factors and cytokine profile as ILC3 [11]. Therefore, we also investigated the Th17 population in the small intestine using a flow cytometer. There was a reduction in the frequency of Th17 cells in the HFD-fed groups compared to the SD control group, with no significant difference between the WT+HFD and IL-1R1^−/−^+HFD groups (Figure 3D,F). Similarly, a decrease in the absolute number of Th17 was noted in the WT+HFD and IL-1R1^−/−^+HFD groups compared to the WT+SD group, though the difference was not statistically significant (Figure 3E).

Considering the reduction of ILC3 in the small intestine, we investigated the expression of cytokines like IL-22 and IL-17, which are usually produced by these cells. The relative expression of Il17a was significantly reduced in the IL-1R1^−/−^+SD group compared to the WT+SD group, whereas it was increased in the IL-1R1+HFD compared to other groups (Figure 3G). On the other hand, the relative expression of Il22 was significantly reduced in the IL-1R1^−/−^+HFD group compared to the WT +HFD group (Figure 3H).

### 3.4. IL-1R1^−/−^ Mice Displayed IL-22 Reduction, Elevated Intestinal Permeability, and Bacterial Translocation after HFD Feeding

IL-22 is important to the maintenance of gut epithelial integrity and local gut microbiota [5]; thus, parameters related to intestinal homeostasis were measured. After 12 h of fasting, the mice were subjected to the FITC-Dextran assay to assess intestinal permeability. The IL-1R1^−/−^+HFD group showed a significant increase in intestinal permeability compared to the other groups (Figure 4A). Given this increase in intestinal permeability, we also evaluated bacterial translocation to VAT, through the relative expression of the bacterial 16S gene. IL-1R1^−/−^+SD mice showed higher bacterial translocation to VAT compared to WT+SD mice (Figure 4B). Additionally, HFD-fed groups exhibited a significant increase in bacterial translocation compared to their respective control groups, but this was amplified in IL-1R1 receptor deficiency (Figure 4B).

In order to investigate factors related to increased intestinal permeability, the gene expression of the ZO-1 junction protein, one of the factors responsible for maintaining intestinal epithelial integrity, was evaluated. Interestingly, the WT+HFD group showed an increase in the relative gene expression of ZO-1 (Tjp1) in the ileum when compared to its control group, whereas IL-1R1^−/−^+SD mice showed a significant reduction in this expression compared to the WT+SD group. At the same time, the IL-1R1^−/−^+HFD group exhibited a significant reduction in the gene expression of ZO-1 compared to the WT group fed an HFD (Figure 4C). Additionally, Reg3g gene expression, which is related to the antimicrobial peptide Reg3γ, was lower in IL-1R1-deficient mice fed an SD or HFD, although the difference was not statistically significant (Figure 4D).

Finally, in order to better understand the profile of the gut microbiota, we detected the phylum and species abundance between experimental groups. An increase in the abundance of the Bacillota (Firmicutes), Pseudomonadota (Proteobacteria), and Verrucomicrobiota (Verrucomicrobia) was observed in the IL-1R1^−/−^+HFD group compared to other groups (Figure 4E). A ratio between Bacillota and Bacteroidota, showed by Firmicutes/Bacteroidetes ratio, was also calculated, as the literature data show that this ratio is often increased in obesity. In this context, we observed that the IL-1R1^−/−^+HFD group exhibited an increase in this ratio compared to the WT+HFD group, indicating that IL-R1 receptor deficiency worsens the relationship between these two phyla in the context of obesity (Figure 4F). Finally, we also evaluated bacterial species in the mouse feces. Akkermansia muciniphila, from the Verrucomicrobiota phylum, was significantly increased in IL-1R1 mice, especially in those that received an HFD (Figure 4G). On the other hand, we did not observe a significant difference in Escherichia coli, from the Pseudomonadota phylum (Figure 4H). Given the increase in Bacillota phylum in IL-1R1-deficient mice, we also evaluated the Clostridium leptum and Ruminococcus torques species. The C. leptum species was increased in both groups fed an HFD, but IL-R1 deficiency exacerbated this increased abundance (Figure 4I). In addition, R. torques was significantly increased in IL-1R1^−/−^ mice, but with greater prevalence in the group that received an HFD (Figure 4J).

## 4. Discussion

Obesity is considered a chronic disease, and multifactorial and environmental factors, such as a sedentary lifestyle and nutritionally unbalanced diet, are often determinant elements in its development. Given the impact this condition has on the body, researchers have been making efforts to understand mechanisms associated with its pathophysiology, as well as preventive and therapeutic strategies. In the field of immunology, much attention has been given to the low-grade inflammation detected in obesity and its metabolic repercussions, mainly involving insulin resistance [12].

Among the pro-inflammatory cytokines, IL-1β is a potent inducer of inflammation that is related to the pathophysiology of several autoimmune and inflammatory diseases [13]. On the other hand, IL-1β also participates in signaling cascades that help maintain homeostasis in various tissues [14]. In this regard, previous studies have demonstrated that deficiency in the IL-1 receptor (IL-1R1) was capable of increasing fat deposition as well as basal glycemia, both in low and high-fat diet contexts, indicating that IL-1β signaling is an important factor in metabolic control [15,16]. These findings corroborate our study since IL-1R1-deficient mice had a higher accumulation of body fat and were prone to developing severe obesity. IL-1 is also crucial to solve infections through the differentiation and function of immune system cells, as well as in the production of other cytokines [14].

Given its diverse functions, this cytokine is abundantly expressed in the intestine, a multifunctional organ. The intestine allows the absorption of dietary nutrients while simultaneously preventing bacteria translocation and their products from passing into the bloodstream from the intestinal lumen [4]. This selective permeability is only possible due to a fine-tuned adjustment between intestinal epithelial cells, the immune system, and the local gut microbiota. In the intestine, resident leukocytes produce cytokines that impact the production of TJ proteins, which keep enterocytes together to avoid the passage of microorganisms or bacterial products into the blood, and antimicrobial peptides, which control the proliferation of certain intestinal bacteria that are potentially pathogenic [17,18].

The gut microbiota, in turn, promotes colonization resistance to infections by competing against pathogenic microorganisms for nutrients, contributing to the differentiation and survival of intestinal epithelial cells, and participating in the formation of gut-associated lymphoid tissue (GALT) [19]. However, this balance can be easily disrupted in the context of a high-fat diet. Previous studies have shown that diet composition is directly related to the concentration of lipopolysaccharide (LPS) in the blood, with both humans and mice consuming a high-fat diet showing higher serum LPS concentrations, reflecting increased intestinal permeability [20,21,22,23]. Additionally, it has also been found that a higher LPS concentration is related to metabolic alterations associated with obesity [24]. Besides LPS passage, Massier and colleagues showed that during obesity and type 2 diabetes, bacterial translocation to adipose tissue occurred after mice were fed an HFD [2]. Indeed, a high-fat diet increased not only intestinal permeability but also bacterial translocation to other sites, such as VAT, and IL-1R1 deficiency worsened this intestinal permeability and bacterial translocation.

To further explore the role of IL-1 and its receptor IL-1R1 in intestinal homeostasis, we aimed to evaluate the cytokines usually produced through their signaling: IL-22 and IL-17 [25,26]. IL-22 is an essential cytokine for tissue repair and the strengthening of the intestinal barrier, achieved by increasing the production of TJ proteins, antimicrobial peptides, mucins, and proliferation of intestinal epithelial cells (IECs) [27]. Studies have shown that IL-22 deficiency led to increased bacterial translocation in mice, as well as worsening of metabolic parameters associated with obesity [5,9,28]. These findings corroborate our data, as the IL-1R1 receptor seemed to be crucial for IL-22 expression, given that IL-1R1^−/−^ mice, in the context of an HFD, showed a significant reduction in its expression, indicating that IL-1β is fundamental for IL-22 production. Additionally, IL-1R1-deficient mice consuming an HFD diet exhibited exacerbated intestinal permeability and bacterial translocation, concomitant with worsening obesity and glucose intolerance. These data, along with other studies, suggest that IL-1 via IL-1R1 receptor accounts for the expression of IL-22, which acts on intestinal barrier function and control of HFD diet-induced obesity.

IL-17 is also a cytokine known to contribute to maintaining the integrity of the intestinal barrier, as previously found by both our group and Maxwell et al. [29,30]. In this regard, we evaluated the gene expression of IL-17 in the ileum. The IL-1R1^−/−^ mice fed an SD showed lower IL-17 expression compared to WT mice. Surprisingly, the IL-1R1^−/−^+HFD group showed increased IL-17 gene expression compared to other groups. It is known that IL-17 does not exert its effects alone; rather, it modulates and amplifies local/systemic signals in a context-dependent manner, potentially exhibiting a more pathogenic or protective profile, depending on the local microenvironment [31]. In this sense, an IL-1R1 receptor deficiency appears to promote an environment with increased IL-17 gene expression, either for a more pro-inflammatory environment contributing to increased intestinal permeability or for a compensatory action attempting to restore epithelial integrity, as well as to contain gut dysbiosis in the context of obesity [32]. It is important to mention that the increase in IL-17 expression may be driven by IL-23 signaling, as IL-1β is not acting due to the lack of its receptor signaling since both cytokines act synergistically for the production of IL-17 as well as IL-22 [10].

Since the association between reduced TJ proteins and intestinal permeability is well-established, we investigated whether increased permeability would be due to reduced expression of the ZO-1 in the ileum. Previous studies have shown that low ZO-1 expression was associated not only with a more permissive epithelium but also with worsening obesity [33]. Unexpectedly, mice fed an HFD showed elevated ZO-1 expression compared to their SD controls. Furthermore, IL-1R1^−/−^ mice fed an HFD showed reduced ZO-1 expression compared to the WT+HFD group. We also evaluated the populations of Th17 and ILC3 cells, the main sources of IL-22 in the ileum. In this case, HFD alone seemed to influence the frequency of these cells, but IL-1R1 deficiency caused an even greater reduction, consistent with lower IL-22 expression compared to the WT+HFD group. This data indicates that IL-1 signaling via IL-1R1 is important for the induction or function of these cells, especially ILC3 cells. Although there was a tendency for a reduction in Th17 cells in the IL-1R1+HFD group, the result was not statistically significant. Despite the literature showing that Th17 cells produce IL-22, their contribution is more subtle compared to ILC3s, which are the main producers of IL-22 in the intestinal environment [27,34]. These data indicate that IL-1R1 deficiency leads to a reduction in IL-22 and alterations on the intestinal barrier, as a result of the reduced population of ILC3 cells in the small intestine.

The gut microbiota has been extensively studied recently due to its interactions with host metabolism and immune system modulation. The microbiota-host interaction occurs both directly, where immune cells recognize and respond to microorganisms through cytokines and antimicrobial peptide production, and indirectly, through the absorption and distribution of their metabolites. These metabolites, such as short-chain fatty acids (SCFAs) like acetate, butyrate, and propionate, can have local or systemic effects. Previous studies have shown that SCFAs can influence the proliferation of ILC3 cells in the colon, as well as impact their production of IL-22 [35]. Additionally, certain gut bacteria induce myeloid cell cytokine production, such as IL-23 and IL-1β, which are crucial for IL-22 production by ILC3 [7,35]. On the other hand, ILC3s are vital for preserving the balance of gut microbiota by managing physical, chemical, and immune defenses. They secrete IL-22 that interacts with intestinal epithelial cells (IECs) to regulate gut microbiota interactions and ensure bacterial containment [36]. ILC3s also shape gut microbiota composition through the production of IL-22 and lymphotoxin alpha (LTα), affecting populations like segmented filamentous bacteria (SFB) and Lactobacilli [37]. Furthermore, they influence immune responses by negatively selecting microbiota-specific CD4+ T cells and fostering the development of T regulatory (Treg) cells, as well as regulating IgA production, maintaining microbial equilibrium, and preventing gut dysbiosis [38,39].

The gut microbiota composition is directly related to the development of diseases such as obesity, type 2 diabetes, and metabolic syndrome, influencing energy expenditure, energy absorption, and inflammatory aspects [2,40]. Studies have linked obesity and metabolic changes with increases in the Bacillota and Pseudomonadota phyla [41]. We observed an increase in the Bacillota and Pseudomonadota phyla in IL-1R1^−/−^ mice that received an HFD, which was consistent with worsening obesity in this group. Additionally, the Ruminococcus torques and Clostridium leptum species, both from the Bacillota phylum, increased in IL-1R1^−/−^ mice fed an HFD, while Escherichia coli (Pseudomonadota phylum) showed no difference. Although Akkermansia muciniphila is associated with improvements in obesity and insulin resistance [42], we observed an unexpected increase in this species in both WT and IL-1R1-deficient mice that developed obesity. Given the complexity of the interaction between the gut microbiota and the host and the often-conflicting scientific findings, further studies are needed to observe the impact of IL-1R1 receptor deficiency on the composition of the gut microbiota.

In conclusion, we found that IL-1R1 is associated with a reduction in ILC3 cells in the small intestine, which may result in reduced IL-22 expression, leading to a worsening of obesity and MS due to increased intestinal permeability and bacterial translocation to the VAT. Thus, the cytokine IL-1, acting on its IL-1R1 receptor, plays an important role in mitigating obesity and MS, and understanding this pathway is important for the search for new therapeutic tools for obesity and MS. On the other hand, new studies are needed to assess the influence of the IL-1R1 receptor in other tissues in order to verify the systemic impact of its deficiency in the context of obesity.

## Figures and Tables

**Figure 1 microorganisms-12-01717-f001:**
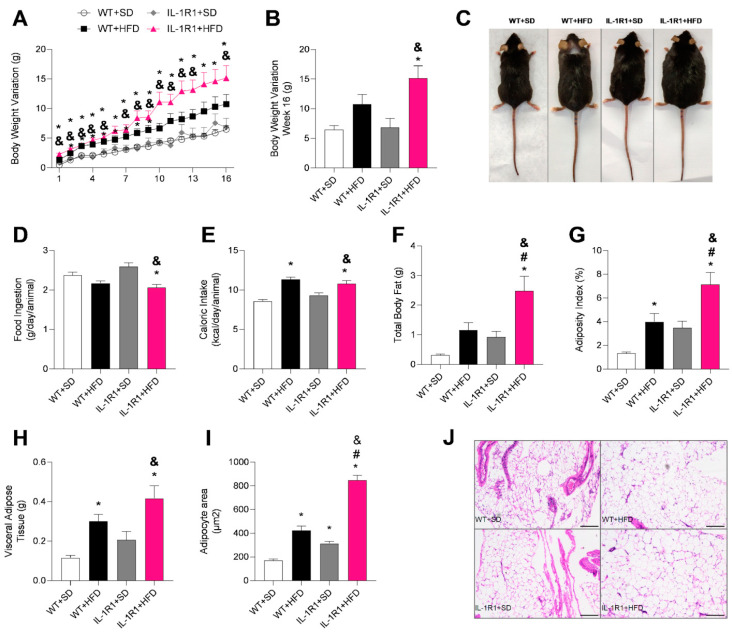
Nutritional and obesity parameters of WT and IL-1R1^−/−^ mice fed a standard control diet (SD) or high-fat diet (HFD) for 16 weeks. Body weight variation was calculated by subtracting the body weight of each week from the body weight at week 0 (**A**); Body weight variation during 16th week (**B**); Representative of weight gain (**C**); Food ingestion (**D**); Caloric intake (**E**); Total body fat (**F**); Adiposity Index (calculated by dividing body fat by body weight and multiplying by 100) (**G**); visceral adipose tissue (VAT) (**H**); adipocyte area from VAT (**I**); VAT histology (at 10× magnification, scale bar = 50 µm) (**J**). Results are represented as mean ± SEM (n = 6 to 12 mice per group). Data were analyzed using the one-way ANOVA test and Tukey’s post-test. * *p* < 0.05 compared to the WT+SD group; ^#^
*p* < 0.05 compared to the WT+HFD group and ^&^
*p* < 0.05 compared to the IL-1R1^−/−^+SD group.

**Figure 2 microorganisms-12-01717-f002:**
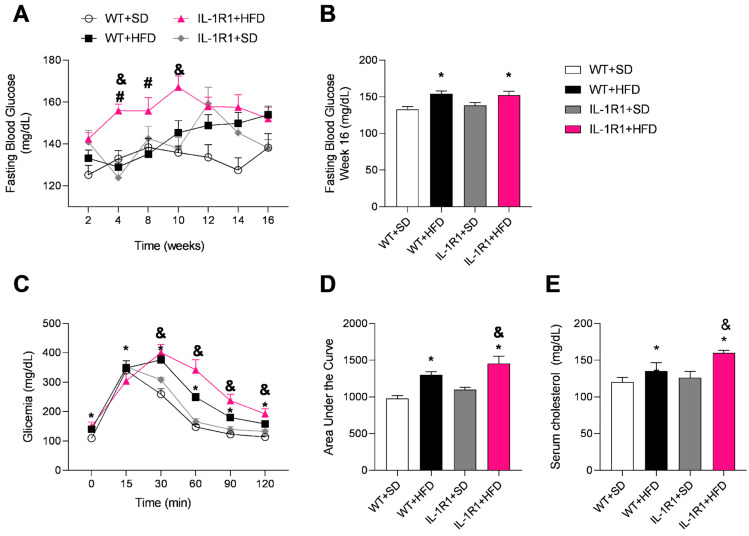
Metabolic parameters of WT and IL-1R1^−/−^ mice fed a standard control diet (SD) or high-fat diet (HFD) for 16 weeks. Fasting blood glucose during the weeks (**A**); fasting blood glucose during 16th week (**B**); Glucose Tolerance Test (GTT) (**C**); Area under the curve from GTT (**D**); Serum cholesterol (**E**). Results are represented as mean ± SEM (n = 6 to 12 mice per group). Data were analyzed using the one-way ANOVA test and Tukey’s post-test. * *p* < 0.05 compared to the WT+SD group; ^#^
*p* < 0.05 compared to the WT+HFD group and ^&^
*p* < 0.05 compared to the IL-1R1^−/−^+SD group.

**Figure 3 microorganisms-12-01717-f003:**
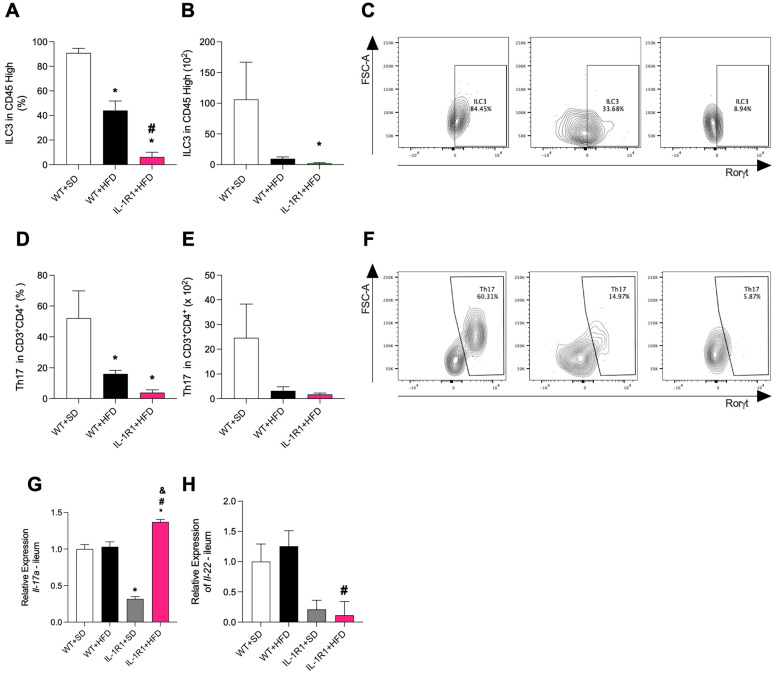
ILC3 and Th17 population in the lamina propria of small intestine in WT and IL-1R1^−/−^ mice fed a standard control diet (SD) or high-fat diet (HFD) for 16 weeks. The small intestine of WT and IL-1R1^−/−^ mice was collected for quantification of the percentage (**A**) and absolute number (**B**) of ILC3 cells. Representative of flow cytometry of ILC3 (**C**). The percentage (**D**) and absolute number (**E**) of Th17 were. Representative of flow cytometry of Th17 (**F**). Fold change in the relative gene expression of IL-17 (**G**) and IL-22 (**H**) in the ileum. Results are represented as mean ± SEM (n = 6). Data were analyzed using the one-way ANOVA test and Tukey’s post-test. * *p* < 0.05 compared to the WT+SD group; ^#^
*p* < 0.05 compared to the WT+HFD group and ^&^
*p* < 0.05 compared to the IL-1R1^−/−^+SD group.

**Figure 4 microorganisms-12-01717-f004:**
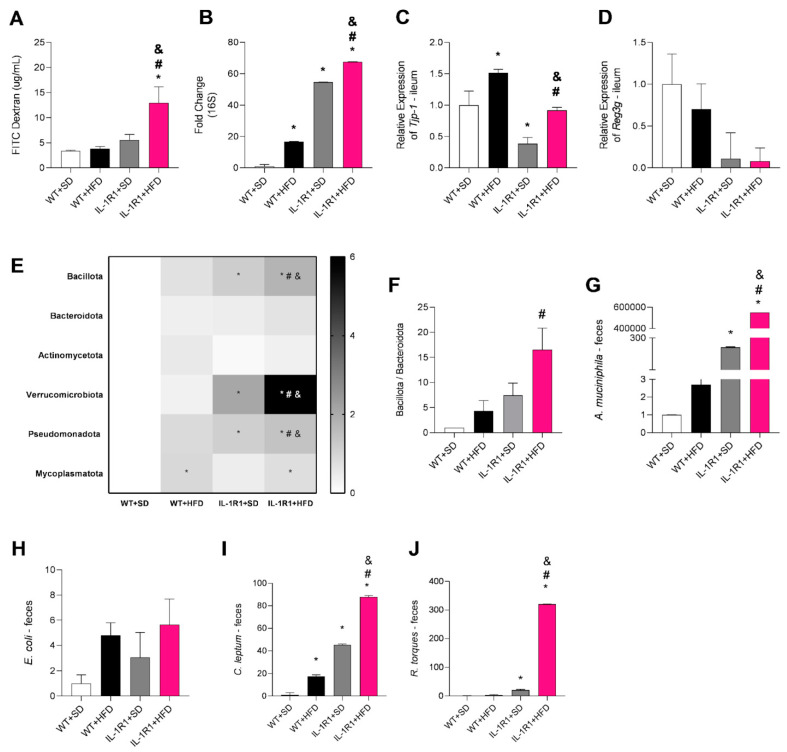
Intestinal permeability, bacterial translocation, and gut microbiota profile of WT and IL-1R1^−/−^ mice fed a standard control diet (SD) or high-fat diet (HFD) for 16 weeks. Quantification of intestinal permeability by FITC-Dextran (**A**); Detection of bacterial 16S gene in VAT by qPCR (**B**); relative gene expression of the tight junction protein ZO-1 (Tjp-1) (**C**) and the antimicrobial peptide REG3g (**D**); heatmap of fold change in the relative abundance of the phyla Bacillota (Firmicutes), Bacteroidota (Bacteroidetes), Actinomycetota (Actinobacteria), Verrucomicrobiota (Verrucomicrobia), Pseudomonadota (Proteobacteria), Mycoplasmatota (Tenericutes) phyla (**E**); Bacillota/Bacteroidota ratio in feces (**F**); fold change in the relative abundance of *A. muciniphila* (**G**), *E. coli* (**H**), *C. leptum* (**I**) and *R. torques* (**J**) in feces. Results are represented as mean ± SEM (n = 6 to 12 mice per group). Data were analyzed using the one-way ANOVA test and Tukey’s post-test. * *p* < 0.05 compared to the WT+SD group; ^#^
*p* < 0.05 compared to the WT+HFD group and ^&^
*p* < 0.05 compared to the IL-1R1^−/−^+SD group.

## Data Availability

The original contributions presented in the study are included in the article/Appendix A, further inquiries can be directed to the corresponding author.

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
