# Peer review of "IL-1 Receptor Contributes to the Maintenance of the Intestinal Barrier via IL-22 during Obesity and Metabolic Syndrome in Experimental Model"

_microorganisms, 2024, doi:10.3390/microorganisms12081717_

Round 1
Reviewer 1 Report
Comments and Suggestions for Authors
We investigated the association between intestinal mucosal permeability and obesity and metabolic syndrome based on IL-1R. At present, this field has received little attention from researchers and is highly innovative. But I think the author can still improve in the following aspects.
1. Abstract chapter, the author wrote too simple, especially the methods part and the results part. A complete abstract can increase the reader's interest.
2. In Fig1-J, I can't tell what kind of tissue it is. If it is liver tissue, I think its fat content is too high, and liver cells are almost invisible. Please mark more clearly.
3. The authors describe the 16s sequencing of gut microbes in the manuscript. According to my understanding, changes in the intestinal immune microenvironment will affect the species and abundance of intestinal microbes, and different microbial species and abundance will also affect host immune regulation. I suggest that the discussion part of the manuscript should further increase the impact of intestinal microecological changes on the host immune microenvironment, and the author may try to discuss it from the perspective of metabolites of microecology.
Author Response
Reviewer #1
We investigated the association between intestinal mucosal permeability and obesity and metabolic syndrome based on IL-1R. At present, this field has received little attention from researchers and is highly innovative. But I think the author can still improve in the following aspects.
- Abstract chapter, the author wrote too simple, especially the methods part and the results part. A complete abstract can increase the reader's interest.
Author response: Thank you so much for your attention in reviewing our manuscript and for your valuable suggestions. We have revised the abstract to provide a more detailed account of the methods and results. The revised abstract can be found on page 1, lines 24-41.
- In Fig1-J, I can't tell what kind of tissue it is. If it is liver tissue, I think its fat content is too high, and liver cells are almost invisible. Please mark more clearly.
Author response: The tissue in the Fig1-J is VAT (visceral adipose tissue). This information can be found on page 16, lines 654-655. This is also described in the Results section on page 6, lines 248-254.
- The authors describe the 16s sequencing of gut microbes in the manuscript. According to my understanding, changes in the intestinal immune microenvironment will affect the species and abundance of intestinal microbes, and different microbial species and abundance will also affect host immune regulation. I suggest that the discussion part of the manuscript should further increase the impact of intestinal microecological changes on the host immune microenvironment, and the author may try to discuss it from the perspective of metabolites of microecology.
Author response: Thank you for you comments. In response to your suggestion, we have added a discussion (page 11, line 455-475) with two paragraphs discussing the gut microbiota versus the immune system, focusing on ILC3 responses.
Reviewer 2 Report
Comments and Suggestions for Authors
The authors demonstrated that IL-1R1-deficient mice became more obese, and exhibited a reduction in ILC3 cells associated with lower IL-22 gene expression, resulting in increased intestinal permeability, bacterial translocation to visceral adipose tissue and aggravation the glucose intolerance. These data indiccates that IL-1R1 receptor acts in the control of intestinal homeostasis. The present study was well-organized and well-investigated, and will give us a novel mechanism . We have no claim in the present paper.
Author Response
Reviewer #2
- The authors demonstrated that IL-1R1-deficient mice became more obese, and exhibited a reduction in ILC3 cells associated with lower IL-22 gene expression, resulting in increased intestinal permeability, bacterial translocation to visceral adipose tissue and aggravation the glucose intolerance. These data indicates that IL-1R1 receptor acts in the control of intestinal homeostasis. The present study was well-organized and well-investigated, and will give us a novel mechanism . We have no claim in the present paper.
Author response: We sincerely appreciate your attention and the time you have dedicated to reading and reviewing our manuscript.
Round 2
Reviewer 1 Report
Comments and Suggestions for Authors
I think the author has answered my questions sufficiently and the paper deserves to be published